# Chronic Microvascular Complications in Prediabetic States—An Overview

**DOI:** 10.3390/jcm9103289

**Published:** 2020-10-13

**Authors:** Angelika Baranowska-Jurkun, Wojciech Matuszewski, Elżbieta Bandurska-Stankiewicz

**Affiliations:** Clinic of Endocrinology, Diabetology and Internal Medicine, Department of Internal Medicine, School of Medicine, Collegium Medicum, University of Warmia and Mazury, Żołnierska 18, 10-561 Olsztyn, Poland; wmatuszewski82@wp.pl (W.M.); bandurska.endo@gmail.com (E.B.-S.)

**Keywords:** prediabetes, diabetes, microvascular complications

## Abstract

A prediabetic state is a major risk factor for the development of diabetes, and, because of an identical pathophysiological background of both conditions, their prevalence increases parallelly and equally fast. Long-term hyperglycemia is the main cause inducing chronic complications of diabetes, yet the range of glucose levels at which they start has not been yet unequivocally determined. The current data show that chronic microvascular complications of diabetes can be observed in patients with abnormal glucose metabolism in whom glycaemia is higher than optimal but below diagnostic criteria for diabetes. Prediabetes is a heterogenous nosological unit in which particular types are differently characterized and show different correlations with particular kinds of complications. Analysis of the latest research results shows the need to continue studies in a larger population and can imply the need to verify the currently employed criteria of diagnosing diabetes and chronic complications of diabetes in people with prediabetes.

## 1. Definition and Prevalence

Diabetes (diabetes mellitus, DM) comprises a group of metabolic disorders characterized by hyperglycemia resulting from disorders in insulin secretion or action. Chronic hyperglycemia in DM leads to damage, impaired function or insufficiency of various organs, especially eyes, kidneys, nerves, the heart, and blood vessels [1]. There are 463 million diabetes patients, that is 9.3% of the world’s population. It is estimated that, in 2045, there will be 700 million patients (10%) [2].

Prediabetes is the main risk factor for DM. Analogically to the increase of DM prevalence, due to their identical pathophysiological background [3], the prevalence of prediabetes also increases. Prediabetes as a separate nosological unit is characterized by impaired glucose metabolism with glycemia above the optimal value but still below the diagnostic levels for DM.

Complete epidemiological data concerning prediabetes are not available as there are no uniform diagnostic criteria that would be used by all scientific societies (Table 1), and there are no representative data from most of the world’s countries [2,4,5].

In the USA, the National Examination of Health and Nutrition Survey (NHANES), in the years 1999–2002 and 2007–2010, reported an increase in the prevalence of prediabetes, which was higher in women than men; yet prediabetes, as such, appeared more often in men. Interestingly, the prevalence of impaired fasting glycemia (IFG) remained quite stable—increasing from 23.9% to 26.6.%, contrary to the criteria which employ glycated hemoglobin (HbA1c) (American Diabetes Association criteria), according to which the prevalence increased from 9.5% to 17.8% [14]. It is estimated that, in 2018, 33.9% (48.3% people above 65 years of age) adult Americans age 18 or older (88 million) suffered from prediabetes, evaluated on the basis of fasting plasma glucose (FPG) or percentage of HbA1c [15]. The global prevalence of IFG is estimated to be 5%, yet one needs to remember that there are two criteria of diagnosing IFG used [16].

Uniform diagnostic criteria of impaired glucose tolerance (IGT) make it possible to compare its prevalence all over the world (Figure 1). In 2019, almost 374 million (7.5%) people in the world were said to have IGT, while it is estimated that, in 2030, this will be 453 million (8.0%), and, in 2045—548 million (8.6%). It is surmised that, in 2030, prediabetes will concern more than 470 million people all over the world [17].

Long-term hyperglycemia is the main cause of chronic complications of DM, which are divided into microvascular (diabetic kidney disease, diabetic retinopathy, and diabetic neuropathy) [18] and macrovascular (ischemic disease of the heart, brain and lower limbs) [19]. There is a “gap” in the glycemic status between correct values and DM. The current review shows that, in many patients, those glucose levels which are between optimal and indicative of DM can also induce the development of chronic microvascular complications. Prediabetes is a heterogenous nosological unit in which particular types are differently characterized and show different correlations with particular kinds of complications. The explanation of the relationship between prediabetes and chronic microvascular diabetes complications is clinically important, as well as important to public health, as early diagnosis of the disorder could be the basis for early therapeutic interventions that could prevent progression of complications or even cause their resolution.

## 2. Methodology

The aim of the study was to discuss the occurrence of chronic microvascular complications of diabetes in prediabetic states.

### 2.1. Pathophysiology

#### 2.1.1. From Correct Glycaemia Through Prediabetes to Diabetes

In healthy people, the blood glucose level is precisely regulated. FPG usually remains at the level 3.9–5.6 mmol/L (70–101 mg/dL) [20]. All over the world, it was observed that the mean FPG increases. In 2008, the mean world standardized FPG was 5.5 mmol/L (99 mg/dL) in men, was 5.42 mmol/L (97 mg/dL) in women, and increased by 0.07 mmol/L and 0.09 mmol/L per decade, respectively [21]. After a meal, the increases infrequently exceed 3 mmol/L (54 mg/dL) [22]. During the development of diabetes mellitus type 2 (DM2), the homeostasis of FPG and post-load glycaemia becomes disrupted [23].

As was proven in studies analyzing dependencies of repeated tests of glucose levels with insulin secretion and sensitivity to insulin, the development of diabetes is a continuous process. In the Whitehall II study, in people who developed diabetes, increased glucose levels were observed as early as 13 years before the diagnosis, although glucose values seemed precisely regulated within the correct range up to 2–6 years before the diagnosis, until their sudden increase [24]. This was also confirmed in other studies [25,26,27]. Sensitivity to insulin was already lowered 13 years before the onset of DM, while its steep decrease occurred 5 years before the diagnosis. Insulin secretion (β-cell function) remained at the same level for 13 years during the observation period, 3–4 years before the diagnosis there occurred a compensational increase of insulin secretion, while immediately before the diagnosis there was a sudden decrease [23]. These results confirm the hypothesis that insulin resistance emerges a number of years before DM, and the reduction of β–cell function appears already in a prediabetic state [20,28] (Scheme 1).

In addition, Weir presented a multi-stage model of diabetes development [29]. The first stage consists in a long period of insulin resistance which is compensated for by increased secretion of insulin and increased β-cells mass [23]. The second stage brings a stable period of adaptation when β-cells gradually cease to be able to compensate for the increased insulin resistance. This period probably starts when FPG and post-load glycemia are still within norm but not optimal [23,24,29]. A number of abnormalities of the first and second stage occur, thus, before the prediabetic stage. During the early stage of decompensation—i.e., the third stage of DM development—β-cells become incapable of compensating for insulin resistance by secreting more insulin, and the glucose level rapidly increases [23,29]. This period probably starts with the prediabetic state and lasts until the manifestation of DM [29].

#### 2.1.2. IFG and/vs. IGT

People with IFG and IGT manifest abnormalities characteristic of DM2, that is, insulin resistance and β-cells dysfunction [30]. However, patients with isolated IFG differ from patients with isolated IGT by their fasting and 2-h post-load glucose levels and the shape of the curves for glucose levels during oral glucose tolerance test (OGTT). Both groups show resistance to insulin, but the places of abnormalities are different.

FPG values are determined by endogenous glucose production (EGP), which depends mainly on the liver. EGP and the fasting insulin level are employed as insulin resistance markers for the liver and show a strong correlation with FPG [22,23,31]. Early (0–30 min) insulin response during OGTT and insulin secretion in the first phase (0–10 min) during the hyperglycemic clamp and intravenous glucose tolerance test are disturbed in IFG [32]. When the FPG level is above 110 mg/dL (6.1 mmol), insulin secretion in the first phase is heavily impaired. However, late (60–120 min) insulin response during OGTT and the second phase of insulin secretion during the hyperglycemic clamp are within range. The hyperinsulinemic euglycemic clamp was used to perform quantitative assessment of glucose uptake dependent on insulin (it first of all reflected muscle uptake), in subjects with IFG insulin sensitivity, was assessed as correct/almost correct. Despite an initially incorrectly high (0–60 min) increase of plasma glucose level after ingesting glucose, the correct sensitivity of muscles to insulin and the correct second phase of insulin secretion allowed 2-h post-load plasma glucose levels in subjects with IFG to come back to correct values (<140 mg/dl, (7.8 mmol/L)). The reason behind increased EPG in subjects with IFG is not entirely comprehensible. It was proved that the basal glucose production rate in the liver (which is the main indicator of FGP level in DM2 patients) is lowered in people with IFG, in comparison with subjects with normal glucose tolerance (NGT). This observation indicates that a decrease in the glucose clearance rate is the most probable reason behind an increased value of FPG in IFG [33]. Summarizing, IFG is characterized by an increased FPG level, high insulin resistance of the liver, impaired early insulin response, excessive early increase (0–60 min) of plasma glucose level during OGTT, and correct 2-h post-load plasma glucose level, as well as correct insulin sensitivity in skeletal muscles [22,23,31].

At the same time, tests with the hyperinsulinemic euglycemic clamp in patients with IGT showed impaired activity of insulin in reducing glucose levels in the entire body (mainly in muscles). Late insulin response (60–120 min) during OGTT and insulin secretion in the second phase during the hyperglycemic clamp are heavily impaired in subjects with IGT [33]. Concluding, in people with IGT, FPG is correct with small changes of insulin sensitivity in the liver, and early and late phase of insulin secretion is impaired [21,22,31], while the main place of insulin resistance occurs in the muscles [20,22].

The above descriptions show considerable differences in the pathophysiological mechanisms of isolated IFG and isolated IGT, while the clinical significance of these disruptions requires further investigation.

#### 2.1.3. The Mechanism Behind Complications

Pathogenesis of microvascular complications in people with glucose metabolism disorders has not been fully explained yet. Apart from genetic predispositions, some of the proposed mechanisms encompass hyperglycemia-induced changes in the pathway of polyol, hexosamine and protein kinase C (PKC), advanced glocalization of proteins, glomerulus filtration, inducing the production of transformation growth factor (TGF-β), and other harmful growth factors, as well as oxidative stress [3]. Among various factors, it seems that these are hyperglycemia-related mechanisms which have a particularly significant effect on the development of the analyzed complications. Intracellular hyperglycemia was found to be related with activation of four toxic pathways which can lead to damage in tissues: increased inflow through the polyol pathway, creating advanced final products of glycosylation, increased activity of the hexosamine pathway and activation of PKC (Figure 2) [34]. Action through those toxic pathways was documented on the basis of experimental models, as well as in people with DM. However, clinical experience shows that actual damage of tissues leading to complications requires a few years of exposure to uncontrolled glycemia. Thus, it is unclear why some people with glycemia below the diagnostic criteria for DM become prone to “premature” development of complications usually observed in patients with long-term DM. It is possible that particular people differ in their response to various levels of glycaemia in relation to the thresholds triggering these toxic pathways. Another possible option is the multiplication effect, in which simultaneous activation of many mechanisms may make some people especially prone to premature damage [3].

In fact, one toxic pathway including PKC activation combines together an increased level of blood glucose with inducing mechanisms triggering tandem changes in expression of nitric oxide synthase (NOS), vascular epithelial growth factor (VEGF), plasminogen-1 activator inhibitor (PAI-1), transformation growth factor β (TGF-β), reactive forms of oxygen, and nucleus kappa-β factor (NF-κβ) (Figure 2) [34]. Effects of glycemic PKC activation, especially those encompassing vascular and inflammatory pathways, can be the cause of impaired β-cells function, insulin resistance, and microvascular and macrovascular complications in susceptible persons. Thus, the current knowledge suggests that moderately increased blood glucose levels in prediabetes can exert negative effect in susceptible persons [3].

### 2.2. Microvascular Complications

#### 2.2.1. Diabetic Retinopathy

A number of studies confirmed a connection between prediabetes and diabetic retinopathy (DR), yet their conclusions are rather diversified, depending on the employed methods [35]. It is estimated that DR can occur in 8–12% of people with prediabetes. In the Diabetes Prevention Program (DPP), DR was found in 7.9% participants with IGT [36]. The connection between IGT and DR was confirmed in the Funagata study, yet the correlation between IFG and DR was not statistically significant [37]. A similar prevalence of DR (8.1%) was found to occur in people with prediabetes in the Gutenberg Health Study [38]. In the AusDiab study, symptoms of DR were diagnosed in 6.7% of people with IFG and/or IGT [39]. A Swedish study carried out among patients with FPG disorders found symptoms characteristic of DR in 10% [40]. Moreover, studies using confocal microscopy and skin biopsy detect small-fiber neuropathy in patients with IGT [41].

In the analysis of three big studies scrutinizing the effect of FPG levels increasing to DM diagnostic values (126 mg/dL) on DR occurrence, it was determined that there is a direct correlation, yet no cohesive and clear threshold, for glycaemia at which DR appears was found [39]. Assessing the connection of DR prevalence with HbA1c and FPG in the population above 40 years of age in the USA, a rapid growth of DR was detected, when HbA1c exceeded 5.5%, and FPG exceeded 104.4 mg/dL, while HbA1c was a better predicative factor than FPG [42]. In a 2019 study among patients with prediabetes (IFG + HbA1c 5.7–6.4%), DR was diagnosed in 24.1% of the group. The percentage of people with DR increased linearly together with the increasing percentage of HbA1c, respectively: HbA1c < 5.7%: DR 11.4%, HbA1c 5.7–6.0%: DR 23%, HbA1c 6.0–6.4%: DR 28.1% [43].

The studies suggest that the effect of glucose and blood pressure on microvascular circulation in the retina is gradual and continuous, and the currently applied definitions of diabetic and hypertensive retinopathy are arbitrary and do not focus on the early stages of the diseases [44]. There are attempts made to assess early symptoms of secondary arteries dysfunction of retina in prediabetes with the use of advanced tools, which cannot be detected in a routine clinical study [45].

#### 2.2.2. Diabetic Kidney Disease

For some years, evidence for a correlation between an increased risk of chronic kidney disease (CKD) and diabetic kidney disease (DKD) in prediabetes has been recorded [35,46]. Characteristic features found in 14 cohort studies are summarized in Table 2 [47,48,49,50,51,52,53,54,55,56,57,58,59,60]. Half of the studies [48,49,50,51,52,53,54] were designed mainly to assess the relationship between metabolic syndrome and CKD. The size of groups in the presented cohorts ranged from 2398 to 118,924. The European population [55,56,58] and the American population [47,48,50] were each assessed in three studies, while eight studies concerned the Asian population [49,51,52,53,54,57,59,60]. Some of the European and American studies included multi-ethnic populations, but the white population constituted at least 50% in each of them. One of the studies concerned only men [52], while others analyzed both sexes. The age of the subjects ranged from 18 to 84 years. The period of observation ranged from 2 to 9 years, with the mean value of 5.5 years. The definition of prediabetes varied in the studies: eight studies considered only IFG, and IFG + IGT, IFG + HbA1c, and IFG + IGT + HbA1c combinations were considered in two studies each. In nine studies, IFG was defined as fasting glucose between 110 and 125 mg/dL; in four studies, this was 100–125 mg/dL, while, in one study, two ranges were taken into consideration [57]. In all the studies scrutinized here, people with diabetes were excluded from participation in the research. In eight studies, estimated glomerular filtration rate (eGFR) < 60 mL/min per 1.73 m^2^ was assumed when defining CKD in line with Modification of Diet in Renal Disease (MDRD); in four studies, Chronic Kidney Disease Epidemiology Collaboration (CKD-EPI) was used, and single studies employed Cockroft–Gault equation [50], or isotope dilution mass spectrometry traceable 4-variable Modification of Diet in Renal Disease (IDMS-MDRD study equation) [57]. Five studies additionally included microalbuminuria or proteinuria in the definition of CKD [49,50,53,57,58]. The degree to which the co-variables were adapted differed in particular studies; only six studies included the baseline eGFR [47,52,55,57,59,60]. Only in four out of fourteen studies did IFG not increase the probability of CKD [53,55,59]; in the remaining studies, it was at least slightly increased. The latest research shows that prediabetes, defined through IGT or HbA1c [57,59,60], is a strong independent predicator for CKD.

A number of studies suggest that, in people with no DM, increased glycemia is related to an increased risk of albuminuria, which is believed to be the earliest marker for DKD. Already, in the NHANES, it was proved that the prevalence of microalbuminuria, and then macroalbuminuria, increased proportionally to worsening of glycemia—that is, from normal glycemia (6% microalbuminuria and 0.6% macroalbuminuria), to IFG (10% and 1.1%), undiagnosed diabetes (29% and 3.3%) and diagnosed diabetes (29% and 7.7%) [46]. Results of a prospective cohort study RENIS-T6 suggest that prediabetes plays an independent role in the development of glomerular hyperfiltration and albuminuria [61]. Discoveries of Melsom et al. indicate that there is a common pathophysiological background which leads to hyperfiltration and albuminuria in prediabetes and early stages of diabetes [62]. However, in the Chronic Renal Insufficiency Cohort (CRIC) study, in people with CKD, prediabetes was not related to an increased risk of eGFR decrease, yet it was related to an increased risk of proteinuria development and increased prevalence of cardiovascular diseases [63].

The range of glucose levels below the DM diagnostic threshold which increase the risk of albuminuria is not clear. Korean researchers who performed a cross-sectional study (n = 5202) divided participants on the basis of their FPG into 5 groups: <5.0, 5.0–5.5, 5.6–6.0, 6.1–6.9, and ≥7.0 mmol/L. In these groups, indicators of albuminuria were, respectively: 4.1%, 6.0%, 7.6%, 12.3%, and 23.4%, which shows an increase of the prevalence of albuminuria not only between normal FPG and IFG but also within the range of IFG [64].

Nevertheless, whether prediabetes can be seen as a cause of the development of CKD and albuminuria still remains unclear because other factors, e.g., hypertension, can also exert their impact adding to the effect of prediabetes [65,66].

#### 2.2.3. Diabetic Neuropathy

Diabetic neuropathy (DN) was considered to be a late complication of DM, yet there is more and more evidence showing its association with prediabetes. DN is the most frequently found complication of newly diagnosed DM [35]. In 18–25% of patients, there are changes in nerve function or their electromyograms results indicate that DN developed in the prediabetic stage. Prediabetes occurs in 25–62% of people with idiopathic peripheral neuropathy. Among people with prediabetes, 11–25% have peripheral neuropathy and 13–21% suffer from neuropathic pain [67]. In the MONICA/KORA study, it was shown that peripheral polyneuropathy appeared in 28% of people with DM, 13% of people with IGT, 11.3% of people with IFG, and 7.4% of people with no glucose metabolism disorders. In the same cohort, neuropathic pain was diagnosed in 13.3% of patients with DM, 8.7% of people with IGT, 4.2% of people with IFG, and 1.2% of people with no glucose metabolism disorders [68].

Additionally, there is more and more evidence indicating that, among people with IGT, the prevalence of idiopathic polyneuropathy, painful sensory neuropathy [69,70,71], and small-fiber neuropathy [72,73] is higher, while the motor fibers are less often affected [67,71]. Non-invasive assessment of neurological disorders in people with IGT revealed a greater prevalence of hyperesthesia, as well as hypoesthesia, and a higher threshold of heat detection [74]. Neuropathy in people with IGT seems to assume a milder course than in the case of DM patients because it affects mainly small not large nerve fibers [71]. Analogically to earlier data, there is evidence that the prevalence of DN increases proportionally to the gradient of glycemia—from people with normal glycemia, to people with IFG and IGT, to people with diabetes [75]. These findings suggest that small non-myelinated nerve fibers that convey pain and temperature, as well as regulating autonomic functions, are affected in prediabetes before DM develops fully.

A number of studies have concerned the association between autonomic disfunction and prediabetes [76,77,78,79,80,81,82,83,84,85,86,87,88,89,90,91] (Table 3). A great diversity of results was observed in these studies when it comes to the discussed glucose metabolism disorders, the size of the groups undergoing research, and cardiovascular autonomic neuropathy (CAN) testing methods. Only in four out of 16 studies were both IFG and IGT considered [81,83,87,89], and one study encompassed IFG + IGT [87]. In one study, IFG was defined in line with guidelines of the American Diabetes Association (ADA), from 1997—FPG 110–125 mg/dL (6.1–6.9 mmol/L) [79], while, in six studies, the IFG definition according to the 2003 ADA guidelines was followed—FPG 100–125 mg/dL (5.6–6.9 mmol/L) [80,81,82,83,87,89]. Nine studies followed IGT defined according to the WHO 1999 guidelines with 2-h post-load glucose levels between 140 and 199 mg/dL (7.8–11.0 mmol/L) in OGTT [78,81,84,85,86,87,88,89]. One study took into consideration an IFG-IGT group (defined according to ADA 2003 and WHO 1999 guidelines) [87], while two studies did not differentiate between these prediabetic states [90,91]. Seven studies included a small group (below 200 participants) [76,77,81,84,86,88,90], and six studies encompassed more than 1000 participants [76,79,80,82,83,91]. The size of groups undergoing research ranged from 80 to 9940. CAN (cardiovascular autonomic neuropathy) assessment tests were diversified: from standardized cardiovascular autonomic reflex tests (CARTs) to heart rate variability (HRV) measurements, but standard procedures and referential values related to the age of participants were not always applied [85,86]. Most of the analyzed studies did not provide the percentage of incorrect tests. In three studies, no differences were found between people with prediabetes and participants with NGT [76,77,84], while nine studies provided limited evidence to confirm lowered HRV results. In studies including both IFG and IGT, a tendency was observed that autonomic indicators in IGT were impaired more severely than in IFG [83,89], or, in the IGT + IFG group, in comparison with isolated IFG and IGT [87,92].

## 3. Future Prospects

Prediabetes is a heterogeneous nosological unit that may require the development of new diagnostic criteria. Recent studies suggest that HbA1_c_ should be included in the permanent diagnostic element due to its high specificity and provide modest improvements in risk discrimination for clinical complications [93]. The 1-h PG during the 75-g OGTT appears to be a useful early biomarker of dysglycemia, which is an indicator of prediabetes [94].

The research into new techniques for the diagnosis of microvascular complications in diabetes is still ongoing. Recent studies suggest that the expression of biomolecules, including microRNAs, proteins and metabolites, are specifically altered during the progression of prediabetes and T2DM and their complications. Their omnipresence in body fluids and the proven relationship of changes in their concentrations at the onset of the disease, during the progression and under treatment of carbohydrate metabolism disorders, make biomolecules a potential prognostic, diagnostic and progressive biomarker [95]. Sidorkiewicz et al. performed a baseline comparison of serum-circulating miRNAs in prediabetic individuals, distinguishing between those who later progressed to T2DM and those who did not [96]. Abnormal expression of long noncoding RNA (lncRNA) and mRNA in people with prediabetes has been shown, and it has been proven that lncRNAs are involved in the entire biological process of prediabetes [97,98,99].

Markers are investigated that may help in the diagnosis of complications in prediabetes in the future [100]. There are attempts to evaluate specific particles that can be used for the diagnosis of DKD [101,102] and DN [103]. Researchers proved that diabetic eye disease is not only DR but also damage to the front surface of the eye [41,104] and the formation of specific genetic biomarkers [105].

In order to inhibit the progression of glucose metabolism disorders, the emergence of DM, and microvascular complications, intensive therapeutic management based on behavioral and pharmacological interventions should be implemented [106]. In a study that compared the incidence of microvascular complications in subjects treated with non-pharmacological therapy versus subjects treated with metformin and/or linagliptin, no statistical difference was found [107]. Currently, clinicaltrials.gov has no data on studies assessing the effect of medicinal products on the incidence of microvascular complications in prediabetic states.

In addition to the potential effects of available antihyperglycemic drugs [108,109], it should be remembered that the hyperglycemia present in prediabetes can upregulate markers of chronic inflammation and contribute to the increased production of reactive oxygen species (ROS) that ultimately causes vascular dysfunction. Oxidative stress and inflammation in prediabetic states may become useful therapeutic points in the future, and targeted immunotherapies may prevent the progression of prediabetes to T2DM and the progression of complications [110].

## 4. Conclusions and Clinical Implications

The prevalence of prediabetes as a civilizational disease has been increasing, which is why it is necessary to establish a new set of diagnostic and therapeutic procedures. The current data indicate that there are microvascular complications of diabetes in people with impaired glucose metabolism in whom glycemia exceeds optimal values but does not yet match diagnostic criteria for DM. The kind and prevalence of diagnosed disorders differ depending on the type of prediabetic state, which may be affected by differences in pathophysiological mechanisms of their development. The proper diagnosis of prediabetic state is therefore crucial and should involve assessment of microvascular complications prevalence; thus, it should be based on applying new, more sensitive and specialized diagnostic tools. Due to the constantly growing number of people with prediabetic state, in order to inhibit the progression of glucose metabolism disorders, the development of DM, and microvascular complications, intensive therapeutic management based on behavioral and pharmacological interventions tailored to the patient’s individual needs should be implemented. Data on cardiovascular risk and all-cause mortality in people with microvascular complications are limited, but recent data indicate that pre-diabetes combined with prehypertension, but not prediabetes alone, in subjects without cardiovascular diseases might elevate the risk of all-cause mortality [111]. It is justified to continue research in this field in order to discover glucose thresholds which induce the development of microvascular complications of DM and to verify the currently applied criteria for diagnosing prediabetes and DM.

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
