# Peer review of "Chronic Microvascular Complications in Prediabetic States—An Overview"

_jcm, 2020, doi:10.3390/jcm9103289_

Round 1
Reviewer 1 Report
The manuscripts provides a very nice concept and caveats in current system used to diagnose prediabetes and associated microvascular complications and need for a more appropriate universal system for better and timely diagnosis.
I think there is scope to improve the last paragraph of the introduction section. Although the microvascular complications are discussion in the later sections of the review, it would be better for fresh/new reader to establish a relationship first and then understand the problem in the system. So author can expand a little bit on microvasular complication and its relationship with DM without being too repetitive and then introduce the caveats.

Author Response
Dear Reviewer,
Thank you very much for your constructive review. Detailed comments significantly increased the substantive value of the manuscript.
We have described in more detail the relationship between prediabetes and microvascular complications in order to introduce the reader to the topic and present the main problems - verses 62-67.We changed "current data" on "current review" in verse 60.
Yours faithfully,
Authors

Reviewer 2 Report
It is a very nice review on an important topic. Tables are very instructive; references are updated.
In this review the authors investigated in depth the available evidence on the association between pre-diabetes mellitus and microvascular complications, highlighting an interesting association between them. They provide a very nice description of the available evidence, summarizing it in detailed Tables. References are updated. I offer the following comments: 1) The authors provide a nice description of the current literature on this topic. I suggest to include some clinical implications that can be derived from study findings. 2) 2) Can the authors provide a paragraph on future perspectives on this topic? Can the authors also check on the clinicaltrials.gov whether there any trials currently ongoing investigation pre-DM; and microvascular complications (any trials on preventive therapeutic approach?). This can further support the key messages of this review. 3) Please, can the authors provide a small paragraph briefly presenting data also on the association between pre-DM and macrovascular complication, especially acute myocardial infarction (Marenzi et al, Diabetes Care 2018 in whom 25% of acute myocardial infarction patients had pre-DM). 4) Can the authors provide data on how pre-DM and microvascular complications interact on affecting patient’s prognosis (hard endpoints). 5) A summarizing figure of this review may be of interest.
Author Response
Dear Reviewer,
Thank you very much for your constructive review. Detailed comments significantly increased the substantive value of the manuscript.1) A very apt observation was the suggestion to supplement the article with the clinical implications, which we described in verses 335-353 and particulary in chapter "Future prospects".
2) Similar to the issue of future perspectives. Thank you for your suggestion, we have described them in a new section "Future prospects" in verses 300-333.
4) Data on the prognosis of patients with chronic microvascular complications are limited, but we have included them in the verses 348-351.
3) and 5) Chronic macrovascular complications in prediabetes are a very important issue. Taking into account the considerable extensiveness of the issue, it was very difficult to describe everything in a short paragraph. We are going to prepare a new manuscript thoroughly describing macrovascular complications in prediabetes.
Yours faithfully,
Authors
